# Novel Spiral Silicon Drift Detector with Equal Cathode Ring Gap and Given Surface Electric Fields

**DOI:** 10.3390/mi13101682

**Published:** 2022-10-06

**Authors:** Jiaxiong Sun, Zheng Li, Xiaodan Li, Xinqing Li, Xinyi Cai, Zewen Tan, Manwen Liu, Hongfei Wang

**Affiliations:** 1College of Physics and Optoelectronic Engineering, Ludong University, Yantai 264025, China; 2School of Integrated Circuits, Ludong University, Yantai 264025, China; 3Engineering Research Center of Photodetector Special Chip in Universities of Shandong, Ludong University, Yantai 264025, China; 4School for Optoelectronic Engineering, Zaozhuang University, Zaozhuang 277160, China; 5Institute of Microelectronics, Chinese Academy of Sciences (IMECAS), Beijing 100029, China; 6School of Physics and Physical Engineering, Qufu Normal University, Qufu 273100, China

**Keywords:** silicon drift detector (SDD), spiral ring cathode gap, electric potential, best drift electric field, optimal electron drift channel

## Abstract

Since the advent of semiconductor detectors, they have been developed for several generations, and their performance has been continuously improved. In this paper, we propose a new silicon drift detector structure that is different from the traditional spiral SDD structure that has a gap between the cathode ring and the width of cathode ring, increasing gradually with the increase of the radius of the cathode ring. Our new structure of spiral SDD structure has equal cathode ring gap and a given surface electric field, which has many advantages compared with the traditional structure. The novel SDD structure controllably reduces the area of silicon oxide between the spiral rings, which in turn reduces the surface leakage current due to the reduction of total oxide charge in the silicon oxide and electronic states on the silicon/silicon oxide interface. Moreover, it has better controllability to adjust this spiral ring cathode gap to achieve better surface electric field distribution, thus realizing the optimal carrier drift electric field and achieving the optimal detector performance. In order to verify this theory, we have modeled this new structure and simulated its electrical properties using the Sentaurus TCAD tool. We have also analyzed and compared different spiral ring cathode gap structures (from 10 µm to 25 µm for the gap). According to the simulation results of potential, electric field, and electron concentration, we have obtained that a spiral ring cathode gap of 10 µm has the best electrical characteristics, more uniform distribution of potential and surface electric field, and a more smooth and straight electron drift channel.

## 1. Introduction

Since the concept of silicon drift detector (SDD) was first proposed by E. Gatti and P. Rehak in the 1980s [1], silicon drift detectors have been widely used in physics frontier fields, such as the Higgs boson particle at the Large Hadron Collider (LHC) [2,3,4], nuclear physics and photon [5,6], dark matter detection [7], X-ray fluorescence spectrometer [8,9,10], medical imaging [11,12] and X-ray pulsar navigation [13,14,15]. For example, the pulsar X-rays navigation technology is applicable in deep space exploration and interstellar flight missions, which provides a new idea and realization way for autonomous navigation of navigation satellites. National Aeronautics and Space Administration (NASA) Marshall Space Flight Center has been collaborating with Brookhaven National Laboratory (BNL) to develop a modular SDD X-ray spectrometer (XRS) intended for fine surface mapping of the light elements of the moon [16]. The traditional silicon drift detector has developed from silicon strip detectors [17,18] to concentric ring SDD and spiral SDD [19], which solves the problem that the collection anode (or cathode) area of silicon strip detectors increases with the increase of detector area, resulting in an increase in the detector capacitance. The anode of the concentric ring SDD is designed in the center of the front side of the detector with concentric cathode rings surrounding the anode [20]. However, the concentric ring structure cannot realize automatic voltage dividing [21]. It is necessary to manually add different voltages to cathode rings to form a reasonable potential gradient. This voltage division method works as a resistive chain that cannot achieve the required nonlinear distribution of surface potential, making the required resistive chain difficult to select. In response to this problem, BNL developed a spiral bias adapter system with silicon drift detector (SBA-SDD) in 2013. But when it is applied to large area SDD or SDD array [22], the potential and electric field of the detector will be relatively high, resulting in an increase in the resistance of the voltage divider, and therefore high-power consumption, heating-up, short circuit, and other problems. Spiral SDD is one of the major detector structures in the SDD family, and is the mainstream of modern international technology development. Most of the structural designs of spiral SDDs are designed and modeled around carrier drift channels. As shown in Figure 1, reference [23] refers to how to calculate the carrier optimal electron drift channel according to its physical model, resulting in a spiral calculation to determine the ring width, ring gap, surface electric field distributions, and other technical parameters. As a continuous resistance chain, spiral ring has the mechanism of automatic voltage distribution, which avoids many difficulties in structural design and complex problems in the manufacture process. The resistance distribution of the spiral ring is related to the ion implantation concentration, ring width, and pitch of the spiral ring itself. In traditional spiral SDD, as the spiral pitch increase with radius, the gap between the cathode rings will also increase.

To optimize the drift behavior of carriers in SDDs and reduce their surface leakage current, we propose a new detector structure based on controlling spiral ring cathode gap, namely, to keep it small and unchanged. Relative to the traditional design that the gap gradually increases with ring radius, this new structure will greatly reduce the silicon oxide area. This in turn can greatly reduce surface leakage current caused by the electronic states in silicon oxide and on the silicon/silicon oxide interface. At the same time, in the design, we can reasonably adjust the given surface electric field according to the actual application to achieve better carrier drift electric field and minimize surface current of the detector, so as to form a fully depleted detector region and realize a high-quality SDD. In this paper, simulations and performance comparisons are performed using the technical computer aided design (TCAD) tool for spiral ring cathodes with equal gaps of 10 µm and 25 µm, respectively, for a given surface electric field. Systematic comparative analysis can also be performed for surface electric fields in subsequent work.

## 2. Detector Structure and Design

Considering the dead space and the symmetry of the physical structure of the detector when it is made into an array, we chose a double-sided hexagonal detector structure in our case of simulation. The shape of the detector is hexagonal, with a diagonal length of 3000 µm and a thickness of 300 µm. The detector bulk is lightly doped N-type silicon substrate with a doping concentration of $4\times 10^{11}\;/{\mathrm{cm}}^{3}$. Figure 2 and Figure 3 show the front view (or top view) and the X=0 (Y−Z positive plane) cross section of the detector respectively. A collection anode is placed in the center of the front surface of the detector, heavily doped with a doping concentration of $1\times 10^{19}\;/{\mathrm{cm}}^{3}$ of N-type and a doping depth of 1 µm. A closed cathode ring is designed just outside the anode, which plays a buffer role, automatically adjusting the voltage between the anode and the spiral cathode ring, and makes the electric field distribution more uniform. The cathode ring is surrounded by a spiral cathode ring that extends outward along the hexagonal trajectory. A protective ring outside of the spiral ring is designed to form a part of the outermost boundary to reduce the boundary high field effect. The closed cathode ring, the spiral cathode ring, and the protective ring are all with P-type heavy doping of a concentration of $1\times 10^{19}\;/{\mathrm{cm}}^{3}$, and a doping depth of 1 µm. Figure 4 shows the backside of the detector. There is no anode on the backside and the cathode on the backside is composed of a cathode disk and a spiral ring system. The number and shape of spiral rings on the backside are the same as those on the front side, forming a completely symmetrical double-sided spiral ring structure. The front side contacts are placed on four electrodes: the anode, the closed cathode ring, the innermost starting position of the spiral ring, and the protection ring. The backside contacts are placed on two electrodes: the disk cathode (the innermost starting position of the spiral ring) and the protection ring. The contact for each electrode is covered with an aluminum layer of a thickness of 1 µm [24]. Silicon dioxide layers with a depth of 0.5 µm are placed on both sides of the detector where there is no implanted electrode.

An appropriate potential gradient is established for the carrier drifting from the position of incident particle to the collection anode by adjusting the cathode gap effectively and controllably to divide the voltage and to fully deplete the detector. Given the surface electric fields on both surfaces, the depletion region within SDD forms a transverse drift electric field that generates a drift channel. When the incident particle induced free carriers enter the drift channel, they drift to the collection anode through the transverse drift electric field, converting the energy of the incident particle into an electrical output signal. This output signal may be used to detect or identify the incident particle or light.

Between each neighboring two spiral rings at a radius of r, we have the electric potential difference ΔV(r) as the following:(1)ΔV(r)=E(r)•p(r)
where p(r) is the spiral pitch, and E(r) is the electric field on the front surface, also, from Ohm law, we have:(2)$$\Delta V\left(r\right)=\frac{I\rho_{s}\alpha r}{\omega \left(r\right)}$$
where I is the spiral electrical current, $\rho_{s}$ the sheet resistance, ω(r) the spiral width, and αr (where α is a constant whose value changes according to the spiral geometry [23], for a hexagonal structure, α=6 is used in this paper) is perimeter of the spiral at r. From Equations (1) and (2) we obtain:(3)$$E\left(r\right)•p\left(r\right)=\frac{I\rho_{s}\alpha r}{\omega \left(r\right)}$$

Here, the helix satisfies the following conditions:(4)p(r)=ω(r)+g
where g is the gap between the implanted area of two neighboring spiral rings, a constant in this new SDD structure. According to our actual situation, we apply a bias voltage value of 110 V on the outermost ring of the spiral cathode ring. To obtain the optimal carrier drift electric field, we chose a given front surface electric field E(r)=f(r). According to Equations (3) and (4), this given electric field E(r) is related to the gap, the spiral pitch p(r), spiral ring cathode width ω(r), implantation sheet resistivity $\rho_{s}$, spiral ring current I, and length per turn αr as the following:(5)$$E\left(r\right)=f\left(r\right)=\frac{\rho_{s}\alpha rI}{p\left(r\right)\left(p\left(r\right)-g\right)}$$

Given the surface electric field and equal (constant) gap, the formula of pitch p(r) can be derived as follows:(6)$$p\left(r\right)=\frac{1}{2}\left[g+\sqrt{g^{2}+\frac{4\rho_{s}\alpha rI}{f\left(r\right)}}\right]$$

The angle φ(r) turned by any point on the spiral ring cathode relative to the starting point can be obtained from Equation (6) and reference [23]:(7)$$\varphi \left(r\right)=4\pi \int_{r_{1}}^{r}\frac{dr}{g+\sqrt{g^{2}+\frac{4\rho_{s}\alpha rI}{f\left(r\right)}}}=\frac{4\pi }{g}\int_{r_{1}}^{r}\frac{dr}{1+\sqrt{1+\frac{\phi_{I}r}{g^{2}f\left(r\right)}}}$$

Here, $\phi_{I}=4\rho_{s}\alpha I$. Let:(8)$$R^{2}\left(r\right)=1+\frac{\phi_{I}r}{g^{2}f\left(r\right)}$$
(9)$$\varphi \left(r\right)=\frac{4\pi }{g}\int_{r_{1}}^{r}\frac{dr}{1+R\left(r\right)}$$

Let:(10)$$R\left(r\right)=\sigma r^{\beta }$$
(11)$$\varphi \left(r\right)=\frac{4\pi }{g}\int_{r_{1}}^{r}\frac{dr}{1+\sigma r^{\beta }}$$

Let:(12)$$y=1+\sigma r^{\beta }$$

Then $r={\left(\frac{y-1}{\sigma }\right)}^{\frac{1}{\beta }}$ and $dr=\frac{1}{\beta \sigma^{\frac{1}{\beta }}}{\left(y-1\right)}^{\frac{1-\beta }{\beta }}dy$. We have:(13)$$\varphi \left(r\right)=\frac{4\pi }{\beta g\sigma^{\frac{1}{\beta }}}\int_{y_{1}}^{y}\frac{{\left(y-1\right)}^{\frac{1-\beta }{\beta }}}{y}dy$$

If $\frac{1-\beta }{\beta }=\mathrm{integer}=j$, Equation (15) can be solved analytically:(14)$$\varphi \left(r\right)=\frac{4\pi }{\beta g\sigma^{\frac{1}{\beta }}}\int_{y_{1}}^{y}\frac{{\left(y-1\right)}^{j}}{y}dy$$

Here, $\beta =\frac{1}{j+1}$.
(15)$$\{\begin{array}{c} j=\cdots ,-4,-3,-2,\;0,1,2,3,\cdots \\ \beta =\cdots ,-\frac{1}{3},-\frac{1}{2},-1,1,\frac{1}{2},\frac{1}{3},\frac{1}{4},\cdots \end{array}$$

Since, y>1, we can rewrite:(16)$$\varphi \left(r\right)=\frac{4\pi }{\beta g\sigma^{\frac{1}{\beta }}}\int_{y_{1}}^{y}\frac{y^{j}{\left(1-\frac{1}{y}\right)}^{j}}{y}dy$$

Since:(17)$$\{\begin{array}{c} {\left(1-\frac{1}{y}\right)}^{j}={\left[\left(-\frac{1}{y}\right)+1\right]}^{j}=1+\frac{j}{1!}\left(-\frac{1}{y}\right)+\frac{j\left(j-1\right)}{2!}{\left(-\frac{1}{y}\right)}^{2}+\\ \cdots +\frac{j\left(j-1\right)\left(j-2\right)\cdots \left(j-i+1\right)}{i!}{\left(-\frac{1}{y}\right)}^{i}+\cdots +{\left(-\frac{1}{y}\right)}^{i} \end{array}\;$$
φ(r) can be solved analytically with integral of each term. Even if j is not an integer, since $\frac{1}{y}<1$, we can use Tylor expansion in Equation (17) to solve Equation (16) approximately with limited terms. From Equations (8) and (10) we have:(18)$$E\left(r\right)=f\left(r\right)=\frac{\phi_{I}r}{g^{2}}\times \frac{1}{R^{2}\left(r\right)-1}=\frac{\phi_{I}r}{g^{2}}\times \frac{1}{\sigma^{2}r^{2\beta }-1}$$

Surface potential ϕ(r) is:
(19)$$\phi \left(r\right)=\int_{r_{1}}^{r}E\left(r\right)dr=\frac{\phi_{I}}{g^{2}}\int_{r_{1}}^{r}\frac{rdr}{\sigma^{2}r^{2\beta }-1}+V_{E1}$$

Let:(20)$$\sigma^{2}r^{2\beta }-1=x$$

Then $r={\left(\frac{x+1}{\sigma^{2}}\right)}^{\frac{1}{2\beta }}=\frac{{\left(x+1\right)}^{\frac{1}{2\beta }}}{\sigma^{\frac{1}{\beta }}}$ and $dr=\frac{1}{2\beta \sigma^{\frac{1}{\beta }}}{\left(x+1\right)}^{\frac{1}{2\beta }-1}dx$. We have:(21)$$\phi \left(r\right)=\frac{\phi_{I}}{g^{2}}\int_{r_{1}}^{r}\frac{1}{2\beta x\sigma^{\frac{2}{\beta }}}{\left(x+1\right)}^{\frac{1}{\beta }-1}dx+V_{E1}$$

If $\frac{1}{\beta }-1=n$ Equation (21) can be solved analytically:(22)$$\beta =\frac{1}{n+1}\left(\begin{array}{c} n=0,1,2,3,\cdots \\ \beta =1,\frac{1}{2},\frac{1}{3},\frac{1}{4},\cdots \end{array}\right)$$
(23)$$\{\begin{array}{c} \phi \left(r\right)=\frac{\phi_{I}}{2\beta g^{2}\sigma^{\frac{2}{\beta }}}\int_{x_{1}}^{x}\frac{{\left(x+1\right)}^{n}}{x}dx+V_{E1}\\ n=\frac{1}{\beta }-1 \end{array}$$

Again, if $\frac{1}{\beta }-1$ is an integer, Equation (21) has an analytical solution and its physical meaning is clear (otherwise it has to be solved approximately through Taylor expansion). Combining Equations (15) and (22), we have:(24)$$\{\begin{array}{c} \beta =1,\frac{1}{2},\frac{1}{3},\frac{1}{4},\cdots ,\frac{1}{m}\\ j=0,1,2,3,\cdots ,m-1\\ n=0,1,2,3,\cdots ,m-1 \end{array}$$
when n is an integer, one can solve Equation (23) with the following results:(25)$$\phi \left(r\right)=\frac{\phi_{I}}{2\beta g^{2}\sigma^{\frac{2}{\beta }}}L\left(x_{1},x,n\right)+V_{E1}$$
where:(26)$$\{\begin{array}{c} L\left(x_{1},x,n\right)=\int_{x_{1}}^{x}\frac{{\left(x+1\right)}^{n}}{x}dx=\int_{x_{1}}^{x}\frac{1}{x}[1+\frac{n}{1!}x+\frac{n\left(n-1\right)}{2!}x^{2}+\cdots \\ +\frac{n\left(n-1\right)\cdots \left(n-i+1\right)}{i!}x^{i}+\cdots +x^{n}]dx\\ =ln\frac{x}{x_{1}}+n\left(x-x_{1}\right)+\frac{n\left(n-1\right)}{4}\left(x^{2}-x_{1}^{2}\right)+\cdots \\ +\frac{n\left(n-1\right)\cdots \left(n-i+1\right)}{ii!}\left(x^{i}-x_{1}^{i}\right)+\cdots \\ +\cdots +\frac{1}{n}\left(x^{n}-x_{1}^{n}\right) \end{array}$$

In this paper, the purpose is reducing the surface current and obtaining the minimum leakage current by effectively and controllably adjusting the cathode gap g to obtain a better drift electric field distribution. From Equations (25) and (26), using the boundary conditions of $\phi \left(r_{1}\right)=V_{E1}$ ($r_{1}$ is the radius of the innermost ring) and $\phi \left(R\right)=V_{Out}$ (R is the radius of the outermost ring), we obtain the following equitation:(27)$$\phi \left(R\right)=V_{out}=\frac{\phi_{I}}{2\beta g^{2}\sigma^{\frac{2}{\beta }}}\left[L\left(x_{1},x_{R},n\right)\right]+V_{E1}$$

One can solved σ:(28)$$\sigma =\{\frac{\phi_{I}}{2\beta g^{2}}L(x_{1},x_{R},n){\frac{1}{V_{out}-V_{E1}}\}}^{\frac{\beta }{2}}$$

For the case of j=0 and β=1, using Equations (12) and (14), we can obtain the spiral ring cathodes as follows:(29)$$r=\frac{1}{\sigma }\left[\left(1+\sigma r_{1}\right)e^{\frac{g\sigma \varphi \left(r\right)}{4\pi }}-1\right]$$

The detector structure used in this paper can be derived from calculations using Equation (29) and related parameters in the design. Figure 2, Figure 3 and Figure 4 clearly show the details related to the SDD hexagonal design, including gaps, dimensions, etc. In our example of calculations, we used n=j=0 and β=1.

## 3. TCAD Simulation Results Analysis

It was mentioned in [23] that the electric field E(r) may appear as a singularity at r=R. Usually, we set an upper limit on the values of $V_{\mathrm{out}}$ and $V_{b}$ to avoid this singularity, and for the two-sided symmetric spiral SDD. In this paper, $V_{\mathrm{out}}$ is generally less than four times the full depletion voltage $V_{\mathrm{fd}}$, here:(30)$$V_{fd}=\frac{qN_{eff}d^{2}}{2\varepsilon_{0}\varepsilon_{Si}}$$
where $N_{eff}$ is the effective doping concentration ($4\times 10^{11}\;/{\mathrm{cm}}^{3}$ here) for N-type light doping of the silicon substrate, q is the electric charge ($q=1.6\times 10^{-19}\;C$), $\varepsilon_{0}$ is the vacuum dielectric constant ($\varepsilon_{0}=8.854\times 10^{-12}\;F/m$), $\varepsilon_{Si}$ is the relative dielectric constant ($\varepsilon_{Si}=11.9$) for silicon [25], and d is the thickness of the silicon substrate (d=300 μm). $V_{out}$ in this case must be large enough to ensure the formation of a suitable drift electric field, especially for the case of this paper where the radius R is very large. The full depletion voltage in this design according to Equation (30) is about 27.3 V. In this paper the detector radius is set at R=3000 μm, the biases at electrode contact points are set at $V_{out}=-84\;v$, $V_{E1}=-6\;v$, $V_{out}^{B}=-75\;v$, $V_{E1}^{B}=-65\;v$.

Using this set of bias voltages, we have simulated this structure using the Sentaurus TCAD tool with results shown in Figure 2 and Figure 4. In order to analyze the effect of different spiral cathode ring gaps on the detector performance, we simulated the structure with a spiral cathode ring gap of 10 µm and a spiral cathode ring gap of 25 µm, respectively, in order to analyze the internal characteristics of the detector for an optimal design. As shown in Figure 3, a cross section is made from X=0 (Y−Z positive plane) in order to better demonstrate the electrical properties of the detector. The corresponding cross section views of potential, electric field, and electron concentration distributions are shown and described below.

### 3.1. Potential Distribution in SDD

Shown in Figure 5 are the two-dimensional potential distributions of detectors in the cross-section of the detector X=0 (Y−Z positive plane) in Figure 3. We can see from the figure that the potential inside the detector is relatively uniform, symmetrically distributed around the anode. Farther away from the central anode, the potential gradually decreases as the radius increases.

The thick black line in Figure 5a is the fitted auxiliary line in the cross section (at Z=30 μm) to obtain a parabola potential as shown in Figure 6. The anode potential is the highest and then decreases to both sides, and the potential in the sensitive region shows a symmetric distribution. According to Figure 5 and Figure 6, we can clearly see a carrier drift path that is marked by the black line. Incident particle or light generated electron carriers (electrons here) in the detector will drift in the potential gradient to the collection anode. Figure 5a shows a more uniform potential distribution than Figure 5b. As a resistor chain, the spiral ring cathode achieves an excellent function of gradual independent voltage dividing, which may improve the detector performance.

### 3.2. Electric Field Distribution in SDD

According to the basic principles of physics, we know that the electric field is the gradient of the electric potential, so we can roughly judge the distribution of the electric field according to the potential distribution in Figure 5.

The entire detector is in a non-zero electric field environment and is completely depleted. From Figure 7 and Figure 8, it can be seen that the electric fields on the front and back sides are not constant. But in the middle of the detector, the electric fields are relatively constant along the drift channel, with a value of about 126 V/cm. The incident particle or light induced electrons gather to the drift channel and then drift rapidly to the anode. Figure 7a shows a more uniform electric field than that shown in Figure 7b. There are electric field in the whole detector bulk in both cases as shown in Figure 7a,b, indicating full depletion achieved in both cases.

### 3.3. Electron Concentration in SDD

As shown in Figure 9, there is a high electron concentration region (yellow colored) in the middle of the detector. This region (yellow colored) is in fact the drift channel of electrons. This drift channel is completely inside the detector, pointing to the collection anode. The electron concentration of this channel is large, but it is still less than the original doping concentration of the substrate itself, indicating that the substrate is completely depleted. Incident particle or light induced electrons at any position of the detector first move to this drift channel driven by electric field, and then drift to the center collection anode driven by a near-constant electric field in the drift channel.

The two-dimensional position resolution sensitivity in the detector is achieved by using the timing of electrons arriving at the collection anode to determine one of the dimensions of position, since the distance of arriving electrons to the collection anode is simply the drift time (timing) times the drift velocity (= electron mobility times drift field $\mu E_{\mathrm{dr}}$). Therefore, a uniform drift electric field is essential to the detector position resolution. It is clear that drift electric field shown in Figure 9a is much better than that in Figure 9b. Therefore, based on all previous data shown in this paper, the SDD with a constant cathode ring gap of 10 µm is much better that the one with a gap of 25 µm.

## 4. Conclusions

In this study, we proposed a new semiconductor detector structure of helical silicon drift detector with equal cathode ring gap and given surface electric field. This new structure reduced detector surface area, which in turn reduced the detector surface leakage current and made the detector internal electric field more uniform. We simulated and analyzed the internal characteristics of the detector with different spiral ring cathode gaps. We obtained the potential, electric field, and electron concentration distribution in order to determine the optimal detector parameters. Simulations were performed to compare two detectors with different spiral ring cathode gaps, and we determined that the performance of the spiral ring cathode gap of 10 µm was superior to the spiral ring cathode gap of 25 µm.

This study provides strong theoretical support for practical fabrication of the detector. The new detector structure can be applied to the fields of space physics and photon science, such as pulsar X-ray detection and X-ray fluorescence spectrometers.

## Figures and Tables

**Figure 1 micromachines-13-01682-f001:**
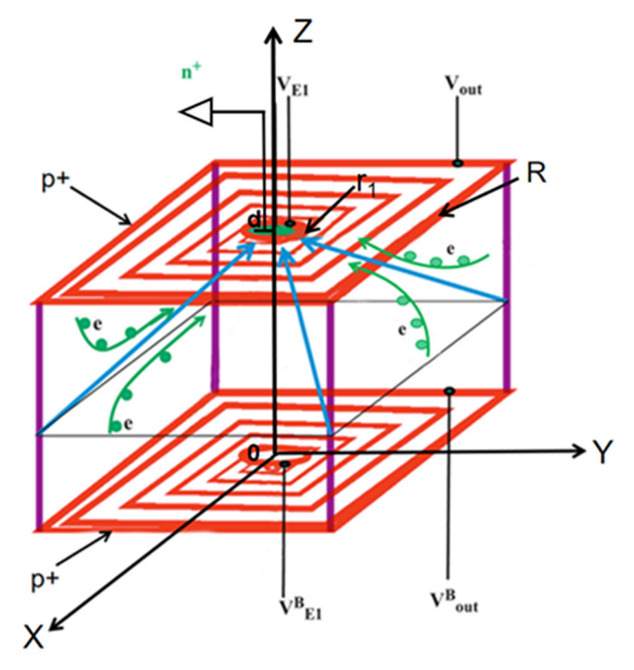
Structure schematic diagram.

**Figure 2 micromachines-13-01682-f002:**
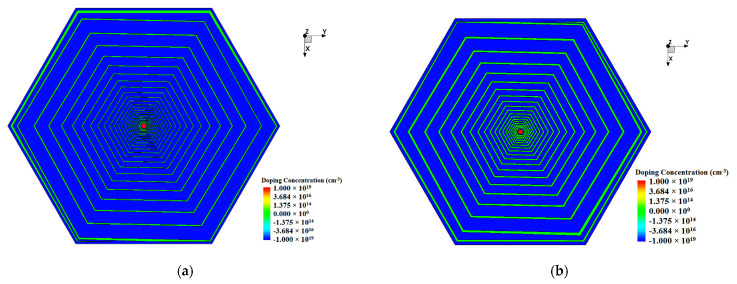
Front side view of the SDD: (**a**) gap 10 µm; (**b**) gap 25 µm.

**Figure 3 micromachines-13-01682-f003:**
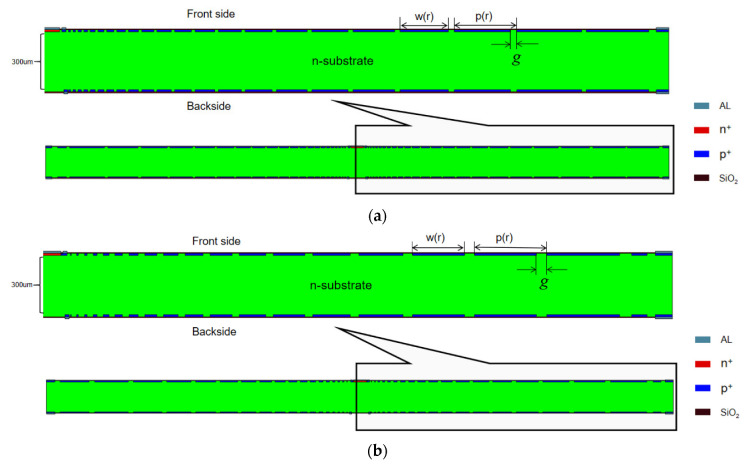
Detector simulation X = 0 (Y-Z positive plane) cross section: (**a**) gap 10 µm; (**b**) gap 25 µm.

**Figure 4 micromachines-13-01682-f004:**
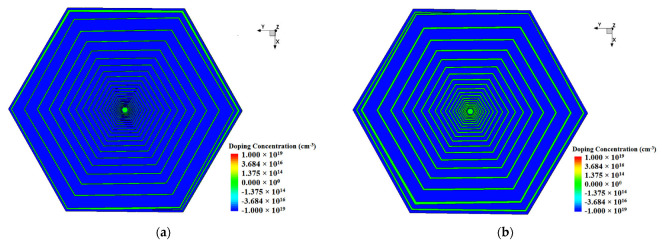
Backside view of the SDD: (**a**) gap 10 µm; (**b**) gap 25 µm.

**Figure 5 micromachines-13-01682-f005:**
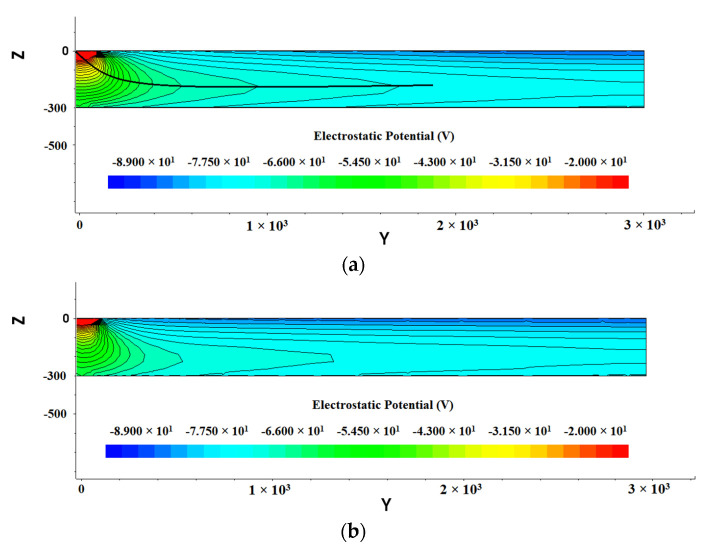
Internal potential distribution of SDD: (**a**) gap 10 µm; (**b**) gap 25 µm.

**Figure 6 micromachines-13-01682-f006:**
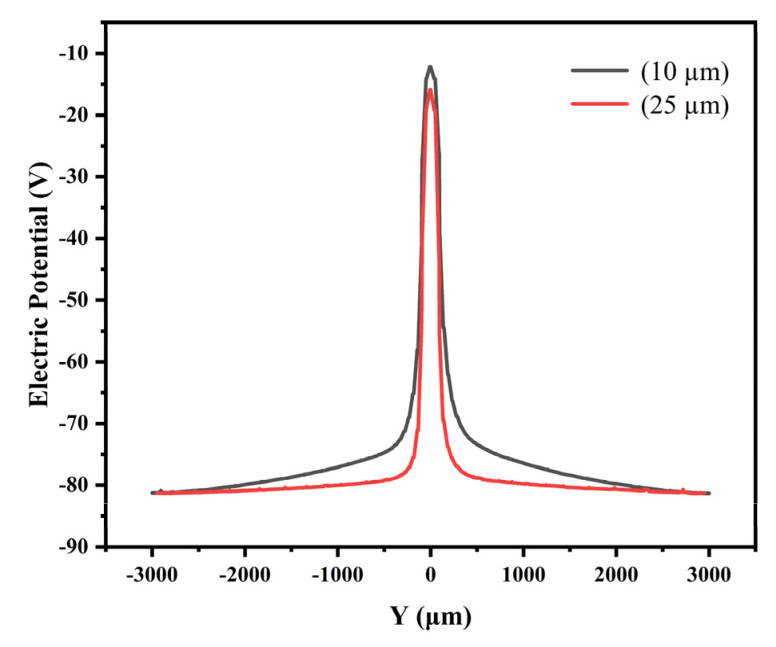
Distribution of potential in one-dimensional section of SDD (Z = 30 μm).

**Figure 7 micromachines-13-01682-f007:**
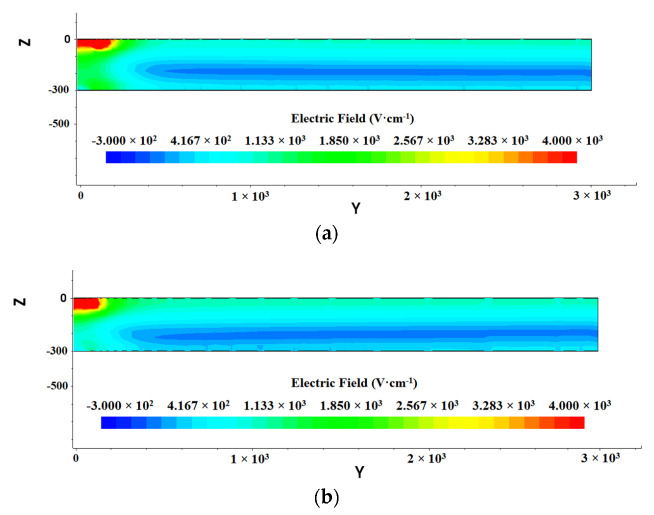
Internal electric field distribution of SDD: (**a**) gap 10 µm; (**b**) gap 25 µm.

**Figure 8 micromachines-13-01682-f008:**
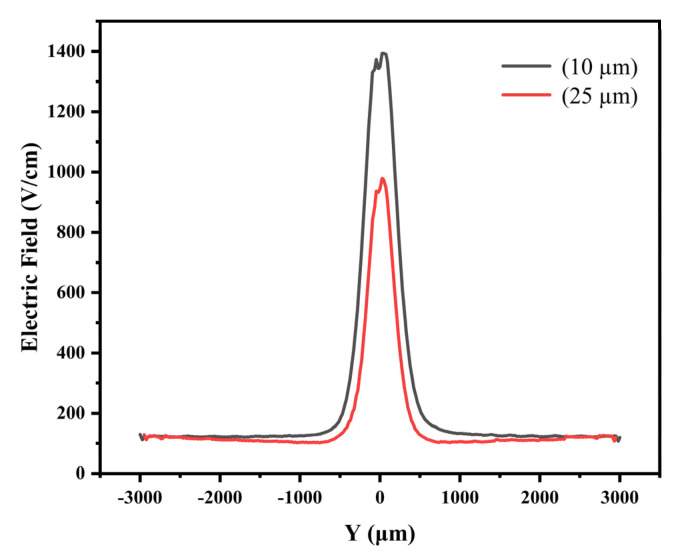
Distribution of electric field in the 1D cross section in SDD (at Z = 210 μm).

**Figure 9 micromachines-13-01682-f009:**
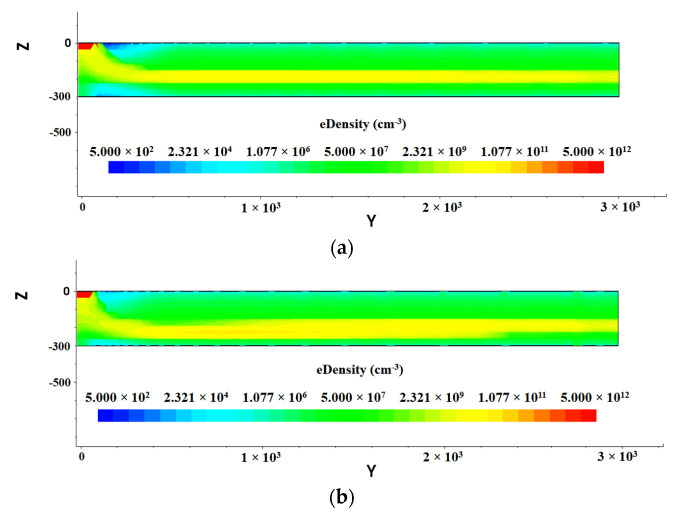
Distribution of electron concentration inside SDD: (**a**) gap 10 µm; (**b**) gap 25 µm.

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
