# Peer review of "Novel Spiral Silicon Drift Detector with Equal Cathode Ring Gap and Given Surface Electric Fields"

_micromachines, 2022, doi:10.3390/mi13101682_

Round 1

Reviewer 1 Report

This manuscript is a well done piece of device design. In my opinion, a few sentences should be added in the introduction to explain to readers, which are not expert, what this device is intended for. 

Once more, in my opinion, the manuscript suffers from a lack of comparison with the performance of an actual detector. I do not know if the construction of the proposed device can rapidly be realized, but I would suggest to see if such a comparison is feasible.

Reviewer 2 Report

The manuscript reports a spiral silicon drift detector (SDD) with an equal cathode ring gap working at a biased voltage. The introduction part provided the background of the research field in a concise way. Then the details of the design procedure were given, and then the electric field of the device was simulated with Sentaurus TCAD software. Two specific cases (with an equal cathode ring gap of 10 μm or 25 μm) were given, and their performance was compared with simulation results.

Despite the topic being relevant to the scientific community and falls into Micromachine’s scope, the concept of the reported SDD device is already published in the authors’ previous publication (ref.15 of the submitted manuscript), and the possibility of having an equal ring gap is also mentioned (see text between eq.32 and eq.33 in ref.15). The modelling and the simulation methods were reported in the authors’ previous publication (ref.15 of the submitted manuscript) in a much more explicitly way. The most significant difference with respect to ref.15 is that two specific cases (gap 10 μm and 25 μm) are given. However, no sufficient discussion on why these two numbers were selected. These two cases solely do not provide much more added value to the scientific community. Therefore, the referee does not suggest this manuscript be published in Micromachines.

Some remarks are reported in the following:

1. The author affiliation part needs to be reformatted according to the requirements of Micromachines or any other journal before resubmission.

2. The acronyms need to be defined the first time they appear. (examples: Page 1/12 LHC, Page 2/12 BNL)

3. Page 4/12 ohm law should be either Ohm’s law or Ohm law.

4. The resolution of the figures needs to be improved. The labels in Fig. 2, 4, 6, 8 are barely visible.

5. Fig. 6 legend missing red line.

6. Fig. 6/8 X-/Y-axis label missing.

7. The authors should be more clear on why the reported two cases (gap 10 μm and 25 μm) were selected, while in principle this number should be proportional to the device radius, according to the ref.15 of the submitted manuscript.

8. Error in eq.5, ‘aIr’ should be ‘arI’?

9 While the Si dielectric constant is usually considered to be 11.7±0.2, the authors used 11.9 in their simulation. Explain why.

Reviewer 3 Report

My first major concern is that the manuscript is not at all suitable for the special issue. 

 Secondly, there are a few errors. In figure 1, the n+ is called anode. In reality, a n-structure can never be called anode however it is biased. 2) line 46 has a reference 13 which is completely unrelated to the claim....

In my opinion, much more comparision with measurements needed to prove this is a better detector.

Round 2

Reviewer 2 Report

All the concerns have been addressed.

Reviewer 3 Report

Accepted for publication